# Integrated Metabolomic and Gut Microbiome Profiles Reveal Postmortem Biomarkers of Fatal Anaphylaxis

**DOI:** 10.3390/ijms26136292

**Published:** 2025-06-29

**Authors:** Yaqin Bai, Zhanpeng Li, Zheng Chen, Li Luo, Jiaqi Wang, Shangman Yao, Keming Yun, Cairong Gao, Xiangjie Guo

**Affiliations:** 1School of Forensic Medicine, Shanxi Medical University, Jinzhong 030606, China; byqlzp@163.com (Y.B.); lzp2058@163.com (Z.L.); 15735558790@163.com (Z.C.); liluo96424@163.com (L.L.); meredithwang@outlook.com (J.W.); yunkeming5142@163.com (K.Y.); 2Health Humanities Research Center, Shanxi Medical University, Jinzhong 030606, China; 15713526812@163.com; 3Key Laboratory of Forensic Toxicology of Ministry of Public Security, Jinzhong 030606, China; 4Translational Medicine Center, Shanxi Medical University, Jinzhong 030606, China

**Keywords:** fatal anaphylaxis, acute myocardial infarction, metabolomics, gut microbiome, biomarker, forensic pathology

## Abstract

The incidence of fatal anaphylaxis is increasing, but there is still no recognized “golden standard” for forensic diagnosis. Due to its non-specific symptoms, especially cardiovascular symptoms without cutaneous changes, it can easily be misdiagnosed as acute myocardial infarction. Here, we established rat models (n = 12) of fatal anaphylaxis (FA), acute myocardial infarction (AMI), and coronary atherosclerosis with anaphylaxis (CAA). The untargeted metabolomics of plasma and 16S rRNA sequencing of fecal matter was performed, and a random forest was used to identify potential biomarkers. Three metabolites (tryptophan, trans-3-indole acrylic acid, and imidazole acetic acid) and three microbial genera (*g_Prevotellaceae_Ga6A1_group*, *g_UCG_008*, and *g_Eubacterium_hallii_group*) were identified as potential biomarkers for distinguishing anaphylaxis and non-anaphylaxis. The classification model of plasma metabolites showed a much better discriminatory performance than that of microbial genus, serum IgE, and tryptase. The performance of the microbial genera was superior to the serum IgE but inferior to the serum tryptase. Forensic samples of fatal anaphylaxis and non-anaphylaxis deaths (n = 12) were collected for untargeted metabolomics detection. The results showed that among the three identified metabolic biomarkers, tryptophan has better stability in cadaveric blood samples. Its diagnostic performance (AUC = 87.1528) was superior to serum IgE and tryptase, making it more suitable as a postmortem biomarker of fatal anaphylaxis.

## 1. Introduction

Anaphylaxis is a serious allergic (hypersensitivity) reaction that can progress rapidly and may cause death [1]. The incidence of anaphylaxis varies worldwide and is often underestimated, but it appears to be increasing [2]. Life-threatening anaphylaxis is characterized by respiratory and/or cardiovascular involvement and may occur without skin/mucosa involvement [1]. Various studies suggest that the heart, particularly the coronary arteries, is often the primary target during anaphylaxis [3]. Acute ischemic events, including angina and myocardial infarction, are now considered to be part of the clinical manifestation of systemic anaphylaxis [4]. This may make the diagnosis of anaphylaxis uncertain. Cutaneous symptoms contribute to distinguishing anaphylaxis from acute myocardial ischemia. However, cutaneous symptoms are absent or unrecognized in up to 20% of patients [5].

Underlying cardiovascular disease (CVD) is a high-risk factor for fatal anaphylaxis [6,7]. CVD also poses a dilemma for forensic pathologists because of the need to distinguish whether the cause of death is fatal anaphylaxis or CVD. Numerous autopsy data have shown that cardiovascular diseases, especially atherosclerotic heart disease, are the most common in fatal anaphylaxis [8,9,10,11,12]. This further confirms that patients with fatal anaphylaxis and underlying cardiovascular diseases are difficult to diagnose accurately and receive effective treatment, which ultimately leads to adverse consequences. However, there are few studies on fatal anaphylaxis and cardiovascular disease, and the association with asthma has been overemphasized [8].

An important issue is the absence of a “gold standard” for diagnosing fatal anaphylaxis. Although measuring various bioactive mediators, such as serum IgE and tryptase, at the time of a reaction can aid in diagnosis [13,14,15], the concentrations of these biomarkers are not altered in all cases. They are related to the course, severity of the reaction, and different allergens [16]. Moreover, many studies have shown that IgE and tryptase are associated with myocardial infarction and death from coronary artery disease in addition to anaphylaxis [17,18,19,20]. Therefore, the diagnosis by these biomarkers has limited sensitivity and specificity.

Metabolomics is a research method reflecting changes in the composition and content of small-molecule metabolites and metabolic pathways in the real-time state of the organism. Metabolomics can detect metabolite changes that occur within minutes (the time range of anaphylaxis) and is an emerging clinical tool to identify new biomarkers for the diagnosis of diseases. In contrast, proteomics and genomic biomarkers typically take longer to be expressed and detected. Several studies have confirmed the metabolic changes in systemic anaphylaxis, depending on anaphylactic triggers, severity, or duration of anaphylactic reactions [21,22,23]. Fan et al. also found that metabolomics can accurately characterize various types of coronary artery disease, including non-obstructive coronary atherosclerosis, angina, and acute myocardial infarction [24]. Another study also demonstrated that differential metabolites can effectively differentiate between anaphylaxis and sudden cardiac death [25]. These findings suggest that metabolomics can reveal metabolic changes in different disease processes and can be used to find novel biomarkers for fatal anaphylaxis. However, previous studies are based on a single anaphylaxis or coronary heart disease, and there is still a lack of research on the differential diagnosis of anaphylaxis in the state of comorbidity.

Accumulating evidence suggests that microbes regulate the host immune system to a certain extent by producing metabolites that affect host metabolism [26]. Diseases and pathological conditions result in dysbiosis of the gut microbiota and altered production of microbial metabolites, leading to dysregulation of the immune system and metabolism. In fact, there is some evidence that intestinal microbiota imbalance and metabolite changes are related to the occurrence and development of allergic diseases, such as food allergy and asthma [27,28,29]. In addition, gut microbial dysbiosis and altered metabolites, such as Trimethylamine N-oxide, are also closely related to cardiovascular diseases, such as coronary atherosclerosis and acute myocardial infarction [30,31]. Importantly, many immunomodulatory bacterial metabolites arise from their metabolism of dietary components (lipids, carbohydrates, and proteins). High-fat-diet-mediated changes in the gut microbiota may increase the risk of food allergy [32], as well as cardiovascular disease [33]. Since this study involved a high-fat diet, we will integrate plasma metabolites with changes in the gut microbiota to investigate their association with anaphylaxis and cardiovascular disease.

In summary, we established single-disease and comorbidity rat models (including fatal anaphylaxis, acute myocardial infarction, and coronary atherosclerosis with anaphylaxis) and detected and integrated rat plasma metabolites and gut microbial profiles to identify specific biomarkers for different disease states and explore related pathways and mechanisms of diseases. Forensic samples of fatal anaphylaxis and non-anaphylaxis deaths were collected to evaluate the application potential of metabolic biomarkers.

## 2. Results

### 2.1. The Characteristics of Rat Models

Forty-eight rats were randomly divided into four groups (n = 12), which were the fatal anaphylaxis group (FA), acute myocardial infarction (AMI) group, coronary atherosclerosis with anaphylaxis (CAA) group, and control (CON) group. Figure 1A shows the schematic diagram of rat model preparation. Within 5 min after the ovalbumin (OVA) challenge, the rats in the FA and CAA groups showed different degrees of anaphylactic symptoms, such as cyanosis, restlessness, scratching, shivering, shortness of breath, defecation, and so on (Figure 1B). In total, 75% of the rats died of fatal anaphylaxis within 40 min to 2 h after OVA challenge. The remaining rats’ anaphylactic symptoms subsided within 40 to 60 min, and they were excluded from further analysis.

The concentration of serum IgE and tryptase was significantly increased in the FA and AMI groups compared with the control group (Figure 1C). However, the serum IgE concentrations have no statistical difference between the CAA and control groups. This indicated that serum IgE was not a good diagnostic indicator for fatal anaphylaxis. The lung tissue was congested and edematous in the FA and CAA groups, the bronchi were significantly constricted, and increased eosinophils were observed (Figure 1D). Multiple organs, such as the myocardium, were congested and edematous in the FA group. Numerous brown tryptase-positive granules were observed in the lungs of the FA, AMI, and CAA groups by immunohistochemical staining (IHC), while the control group was negative (Figure 1D).

After 12 weeks of high-fat emulsion feeding, the serum total cholesterol (TC), triglyceride (TG), low-density lipoprotein (LDL), and high-density lipoprotein (HDL) concentrations were increased in high-fat-fed rats (including the AMI and CAA group) compared with controls (Figure 2A). The atherosclerosis index (AI) of high-fat-fed rats was significantly higher than the controls (Figure 2B), indicating that the high-fat-fed rats had artery atherosclerosis (AI ≥ 3.8). After coronary artery ligation of the AMI group, the myocardium below the ligation site became pale, and the pulsation was significantly weakened (Figure 2C). The electrocardiogram showed a significant decrease in the heart rate of the AMI group after ligation, along with the presence of pathological Q waves, high T waves, and ST-segment elevation (Figure 2D). Notably, the electrocardiograms of the FA and CAA groups exhibited similar ST-segment elevation following the OVA challenge (Figure 2D). It suggests that an electrocardiogram may misdiagnose fatal anaphylaxis as acute myocardial infarction. The hematoxylin and eosin staining (HE) (Figure 2E) and oil red O (Figure 2F) staining showed arterial subintimal lipid deposition, arterial wall thickening, lumen stenosis, endothelial structure disorder, and peripheral inflammatory cell infiltration in high-fat-fed rats. After ligation of the left anterior descending coronary artery (LAD), pathological changes were observed, including degeneration of myocardial cells, unclear structure, necrosis of myocardial contraction bands, and focal hemorrhage. The CAA group had both pathological features of FA (bronchial constriction, eosinophil infiltration, and increased tryptase) and cardiovascular changes in AMI (atherosclerotic changes in arterial endothelium, ischemia of myocardial tissue, and ST-segment elevation), which confirmed the establishment of the pathological superposition model.

### 2.2. Different Metabolomic Profiles of Different Models

To investigate metabolic changes in different diseases, a plasma-untargeted metabolomics analysis was performed using UHPLC-Q Exactive Orbitrap-MS. The total ion chromatograms showed that the plasma metabolic profiles of the CON, FA, AMI, and CAA groups were significantly different (Appendix A). This implies that different diseases lead to distinct alterations in endogenous metabolites, suggesting the potential for identifying metabolic biomarkers for different diseases.

The unsupervised principal component analysis (PCA) (Appendix A) and the supervised partial least squares–discriminant analysis (PLS-DA) (Appendix A) analysis revealed a clear separation among the four groups. To reduce the random error within the groups, further orthogonal partial least squares discrimination analysis (OPLS-DA) analysis was performed, and the samples of the four groups were significantly separated (Figure 3A). These data revealed distinct metabolomics profiles related to different diseases. The clustered quality control (QC) samples indicated the great reproducibility of the experiment. To observe the specific regulation of endogenous metabolites and identify differential metabolites, the data were analyzed and compared in pairs (Appendix A).

Compared with the control group, we identified 50, 32, and 71 endogenous differential metabolites in the FA, AMI, and CAA groups [variable importance in projection (VIP) > 1 and *p* < 0.05], respectively (Appendix A, Figure 3B). In the FA group, palmitoyl sphingomyelin, Erythro-sphingosine 1-phosphate, and Arginine were down-regulated, while the remaining 47 differential metabolites were all up-regulated. The metabolic pathways (MetPA) analysis showed that fatal anaphylaxis was mainly related to arginine biosynthesis, D-glutamine and D-glutamate metabolism, alanine-aspartate and glutamate metabolism, phenylalanine metabolism, citric acid cycle (TCA cycle), as well as arginine and proline metabolism (Figure 3C). In the AMI group, 21 differential metabolites, including 2-amino-1,3,4-octadecanetriol, citric acid, and proline, were up-regulated. In comparison, the levels of 11 differential metabolites, including docosahexaenoic acid, palmitoyl sphingomyelin, and isobutyric acid, were down-regulated. The MetPA analysis showed that AMI was mainly related to phenylalanine tyrosine and tryptophan biosynthesis, linoleic acid metabolism, and phenylalanine metabolism (Figure 3C). In the differential metabolites, we discovered that 13 metabolites were the same and showed similar changes in both FA and AMI groups. After removing these 13 identical metabolites, 37 specific differential metabolites were identified for FA and 19 specific differential metabolites for AMI.

In the CAA group, the identified 71 endogenous differential metabolites included most metabolites of FA and AMI. Among them, 12 metabolites were present in both FA and AMI. Of the remaining 59 differential metabolites, 22 were the specific metabolites for FA and 14 for AMI. Among these metabolites, 4 metabolites (3-indoxyl sulfate, phenylacetylglycine, kynurenine, and deoxycholic acid) were up-regulated in FA but down-regulated in CAA, and 3 metabolites (linoleoyl ethanolamide, linoleic acid, and hexadecanamide) were down-regulated in AMI but up-regulated in CAA. It indicated an interaction between anaphylaxis and atherosclerosis. The MetPA analysis showed CAA was mainly related to phenylalanine tyrosine and tryptophan biosynthesis, arginine biosynthesis, phenylalanine metabolism, D-glutamine and D-glutamate metabolism, as well as alanine aspartate and glutamate metabolism (Figure 3C). All altered metabolites and perturbed metabolic pathways are shown in Figure 4.

### 2.3. Altered Gut Microbiota Composition in Different Models

To detect the changes in gut microbiota in different diseases, feces of rats in each group were collected for 16S rRNA sequencing. Due to the reduced defecation after anaphylaxis, 16S rRNA sequencing was performed on six samples in each group after removing samples of lower quality. The number of amplicon sequence variants (ASVs) in different models was significantly different, indicating that different diseases lead to alterations in gut microbiota (Figure 5A). The top 10 genera with the largest relative abundance were selected for the taxonomy analysis (Figure 5B). The dominant microbiota at the phylum level were *Firmicutes*, *Bacteroidota*, and *Proteobacteria* in these four groups. The relative abundance of *Bacteroidetes* decreased in FA, while the abundance of *Firmicutes* decreased in AMI. Additionally, the abundance of *Proteobacteria* significantly increased in both FA and AMI. However, the abundance of the three phyla in CAA was more similar to the control group. At the genus level, the abundance of *Prevotellaceae_NK3B31_group*, *Clostridia_UCG-014*, and *Ruminococcus* in the other three groups was significantly reduced compared to the control group. The abundance of *UCG-005* and *Muribaculaceae* in both FA and AMI groups was significantly decreased, while the abundance of *Escherichia-Shigella* was significantly increased. Compared with the other three groups, the FA group showed a significant increase in the abundance of *Lachnospiraceae_NK4A136_group* and a significant decrease in the abundance of *Lactobacillus*. In the AMI group, the abundance of *Bacteroides* and *Blautia* was significantly increased, while *Lachnospiraceae_NK4A136_group* and *UCG-005* were significantly decreased. In the CAA group, the abundance of *Lactobacillus* was significantly increased.

The Chao1, ACE, Shannon, and Simpson indices were used to assess alpha diversity. Chao1 and ACE indices represent community richness. Shannon and Simpson indices represent community diversity. It was noticed that AMI had a significant decrease in the community richness of gut microbiota, and there was no difference in community diversity between groups (Figure 5C). Beta diversity was assessed via a PCoA analysis based on Bray–Curtis distances. The principal coordinates analysis (PCoA) result showed a great distance between the four groups, and an analysis of similarities (ANOSIM) analysis showed that the difference in the microbial community between groups was greater than the difference within groups (R > 0.8, *p* < 0.01) (Figure 5D). An linear discriminative analysis effect size (LEfSe) analysis was used to identify significantly differential gut microbiota as the candidate biomarkers (Appendix A). The cladogram showed that the FA group had more significantly differential microbiota than other groups, especially in *Clostridia* (Figure 5E).

### 2.4. The Random Forest Classification Model

Since the diagnostic challenge often lies in distinguishing whether a patient has experienced anaphylaxis, we reclassified all samples into an anaphylaxis group (including FA and CAA groups) and a non-anaphylaxis group (including CON and AMI groups). All the differential metabolites and microbial genera were included after removing duplicates, and their AUC values were calculated. The 55 metabolites and 10 microbial genera with an AUC ≥ 0.7 were selected as candidate signatures (Appendix A). Two-thirds of the samples were used as the training set for a random forest screening of potential biomarkers (Figure 6A). The top three microbial genera (*g_Prevotellaceae_Ga6A1_group*, *g_UCG_008*, and *g_Eubacterium_hallii_group*) and three metabolites (tryptophan, trans-3-Indoleacrylic acid, and imidazole acetic acid) with the highest mean decrease accuracy values were selected to construct the classification model. The diagnostic performance of the model was verified in the remaining 1/3 of the samples, and the AUC values of the individual biomarkers and the combined models are shown in Figure 6B. The classification model of plasma metabolites (AUC = 93.75), microbial genera (AUC = 75.00), or their combination model (AUC = 96.875) showed a much better discriminatory performance than that of serum IgE (AUC = 57.8125), and three metabolites were superior to serum tryptase (AUC = 79.6875). However, the performance of microbial genera is lower than serum tryptase. Microbial genera are considered promising biomarkers due to their non-invasive properties. However, our findings suggest that the diagnostic performance of microbial biomarkers was less than that of metabolites and serum tryptase. This may be because microbes are susceptible to different dietary patterns. In contrast, plasma metabolites are more practical.

### 2.5. The Correlation Between Metabolites and Microbiota

To integrate our findings of plasma metabolites and gut microbiota associated with the development of different diseases, we next performed Spearman correlation analysis and constructed integrated networks to characterize correlations between metabolites and gut microbiota (Figure 7).

In the FA group, *f_Desulfovibrionaceae* and *g_Bacteroides* were the two microorganisms with the highest degree of connectivity with plasma metabolites in the integrated network. They were both strongly correlated with multiple metabolites (|r| ≥ 0.7, *p* < 0.05), which participate in a variety of metabolic pathways, including bile acid metabolism (taurine and taurochenodeoxycholic acid), TCA cycle (malic acid and fumaric acid), arginine biosynthesis (arginine and spermidine), sphingolipid metabolism (sphingosine and sphingosine-1-phosphate), and histidine metabolism (methylimidazoleacetic acid). In addition, *g_Bacteroides* also showed a strong positive correlation with phenylalanine metabolism (hippuric acid and phenylacetylglycine) and tryptophan metabolism (indole, 3-indoxyl sulfate, and kynurenine). Notably, there was a strong negative correlation between *g_Clostridium_sensu_stricto_1* and arginine (r < −0.8, *p* < 0.05). In the AMI group, the most obvious correlation was between the cardiotoxic triethanolamine and four microbiota (*g_UCG-008*, *g_Candidatus_Stoquefichus*, *f_Lachnospiraceae*, and *g_Desulfovibrio)* (r > 0.7, *p* < 0.05). The correlation network in the CAA group showed obvious differences from the single disease. 2,3-dinor prostaglandin E1 was the most connected plasma metabolite in the comorbid group. It was strongly positively correlated with *f_Prevotellaceae*, *f_Ruminococcaceae*, *g_Ruminococcus*, *g_Prevotellaceae_UCG-001*, and *g_Christensenellaceae_R-7_group* (r > 0.7, *p* < 0.05). Of additional concern were tryptophan metabolites (including indole, 3-indoxyl sulfate, and kynurenine), which negatively correlated with *g_Bacteroides* and *g_Lactobacillus* in the CAA group while positively correlated with *g_Bacteroides* in the FA group.

### 2.6. Tryptophan Is a Reliable Biomarker to Distinguish Fatal Anaphylaxis

To verify the validity of the identified biomarkers from the random forest model, human blood samples were collected in the process of forensic autopsy from the Forensic Science Center of Shanxi Medical University for metabolomics detection. Table 1 shows the characteristics of forensic samples. The serum IgE and tryptase showed no statistically significant differences among FA, AMI, and other causes of death. No differences in the levels of trans-3-indoleacrylic acid and imidazole acetic acid were observed in postmortem human blood samples. However, the abundance of tryptophan in fatal anaphylaxis was significantly lower than that in acute myocardial infarction and other causes of death (Figure 8A). The ROC curve showed that the diagnostic performance of tryptophan in distinguishing fatal anaphylaxis from non-anaphylaxis deaths was 87.1528, which was significantly better than that of serum IgE and tryptase (Figure 8B). It suggests that tryptophan is a reliable biomarker for the postmortem diagnosis of fatal anaphylaxis.

## 3. Discussion

The precise diagnosis of FA is a major problem in forensic medicine. The cardiovascular manifestations of systemic anaphylaxis may make an uncertain, even wrong, diagnosis, especially in the case of patients with CVDs such as coronary atherosclerosis. Extensive autopsy evidence has also shown that FA patients often have severe coronary atherosclerosis [9,10,11], which may pose a challenge for accurate diagnosis. Moreover, the underdiagnosis of Kounis syndrome has been more and more noticed in recent years. Given the complexity of anaphylaxis and coronary diseases, we conducted the rat model study in the comorbidity state. In this paper, we highlighted the distinct characteristics of FA compared to AMI by integrating metabolomics and microbiome results.

In this study, we modified the previous model [34] to induce FA in rats, and the incidence of FA was 75%. After the OVA challenge, the rats developed obvious anaphylactic symptoms, such as cyanosis and shortness of breath, and a significant decrease in heart rate. Major pathological changes were observed in the lungs, including congestion, edema, bronchoconstriction, and increased eosinophils. The serum IgE, serum tryptase, and tryptase immunohistochemical staining were used as auxiliary evidence for the success of the model. The serum IgE and tryptase in the FA group were significantly higher than the control, and tryptase-positive granules were seen in the lung tissues by IHC staining, indicating that the FA model was successfully established.

Researchers commonly use the animal model of direct coronary artery ligation to study AMI. However, this model cannot replicate the clinical conditions based on coronary atherosclerosis. The ApoE^−/−^ and LDLR^−/−^ mice are widely used to study atherosclerosis, but the loss of gene function may cause some unexpected changes. In this study, a rat coronary atherosclerosis model was prepared using a high-fat emulsion gavage combined with an intraperitoneal injection of vitamin D3, followed by ligation of the anterior descending branch of the left coronary artery to cause AMI. Due to the poor absorption of cholesterol and strong metabolism of plasma cholesterol, as well as higher plasma HDL and lower LDL in rats, atherosclerosis would not have been easily formed. Intraperitoneal injection of vitamin D3 based on high-fat feeding can significantly increase the concentration of calcium ions in plasma and destroy the arterial endothelium. At the same time, the addition of sodium deoxycholate and propylthiouracil to the high-fat diet can significantly increase the absorption of cholesterol and reduce the metabolism of cholesterol, to elevate the serum total cholesterol content of the rats. After 12 weeks of a high-fat diet, the concentration of serum lipids increased significantly (AI > 3.8), and lipid deposition was observed by oil red O staining, confirming the development of atherosclerosis. The pale myocardium, elevated ST-segment in the electrocardiogram, and the pathological changes in myocardial ischemia in HE staining confirmed the AMI after ligation of the LAD.

In the CAA group, OVA-induced anaphylaxis based on a rat model of coronary atherosclerosis was used to simulate the complexities of patients with both coronary atherosclerosis and FA. Both the pathological features of FA (bronchial constriction, eosinophil infiltration, and increased tryptase) and the cardiovascular changes in AMI (atherosclerotic changes in arterial endothelium, ischemia of myocardial tissue, and ST-segment elevation) confirmed the establishment of the pathological superposition model. Interestingly, serum IgE was increased in both FA and AMI groups but not in the CAA group. However, the mast cell tryptase was positive in all three cases. The link between increased serum IgE and AMI has been confirmed by many studies [18,35]. IgE can promote coronary plaque instability by activating mast cells to release inflammatory factors. One possible reason for the unchanged IgE in the CAA group is that the development of atherosclerosis is associated with IgG and Fcγ receptor (FcγR) [36,37]. When anaphylaxis was induced based on atherosclerosis, an alternative pathway mediated by IgG/FcγR may be preferentially activated instead of the classical IgE/FcεR pathway [38]. Although the IgG pathway has not been fully demonstrated and is controversial in humans, some studies have found that allergen-IgG receptor complexes may be involved in severe anaphylaxis by activating macrophages and/or neutrophils to release PAF and activate mast cells [39,40,41]. Another possible reason is the counteracting effect between Th2-type anaphylaxis and Th1-type atherosclerosis due to a mutually inhibitory Th1/Th2 balance [40]. Mito et al. found a higher Th1-type cellular response (proliferation and IL-2 production), more mast cells in the tracheal mucosa, and lower antigen-specific IgE levels in high-fat diet mice [42]. However, Schröder et al. [43] discovered that short-term high-fat diet feeding resulted in diminished Th1/17 but unchanged Th2 differentiation, offering protective effects in allergic asthma development. Moreover, Jaramillo et al. [18] also found lower allergin-specific IgE (sIgE) in patients with a history of myocardial infarction, and pointed out that the mutual antagonism between Th1/Th2 may make atopy an independent protective factor for MI, which is consistent with our results.

The results of plasma metabolomics indicated that FA, AMI, and CAA shared the same phenylalanine metabolic process. However, a unique metabolic feature of FA is the dysregulation of arginine biosynthesis and metabolism. L-arginine is a semi-essential amino acid metabolized to L-citrulline and nitric oxide (NO) by nitric oxide synthase isoenzymes. NO can cause vasodilation and increased vascular permeability, resulting in the development of anaphylactic shock. The concentration of NO increases 2~3 min after the onset of anaphylactic shock and continues for up to 60 min [44]. Our study identified a depletion of arginine and an increase in citrulline, spermidine, and proline. Consistent with our findings, Zhang et al. [45] also reported a significant decrease in arginine and an increase in spermidine in the serum of milk allergy children. Shi et al. [46] also found that arginine was reduced in penicillin-induced fatal anaphylactic shock. However, Perales-Chorda et al. [21] found an increase in serum arginine and proline in the acute phase (<2 h) and gradually decreased over time (2~4 h) in moderate allergic patients. These results suggest that the level of arginine metabolism after anaphylaxis may be related to the severity of the reaction.

Another metabolic feature of FA is altered energy metabolism dominated by glutamine/glutamate metabolism and the TCA cycle. Glutamine is an important metabolic substrate that helps rapidly proliferating cells meet their requirements for ATP and biosynthetic precursors. Glutamine enters cells via amino acid transport proteins and is converted to glutamate in mitochondria by a deamination reaction catalyzed by glutaminase. Then, glutamate is converted to α-ketoglutarate by various mitochondrial aminotransferases, such as glutamate dehydrogenase, alanine, or aspartate aminotransferase and participates in the TCA cycle, which contributes to the production of ATP [47]. When the allergic reaction occurs, T cells are activated and clonal-expanded after antigen stimulation, and their energy demand is significantly increased, which needs to be met by glutamine and glutamate metabolism [48,49]. Moreover, glutamine deficiency reduces in vitro proliferation of antigen-stimulated rat and human lymphocytes and impairs plasma cell differentiation and immunoglobulin synthesis and secretion [48]. In addition, glutamine and its metabolized glutathione have anti-inflammatory activity, which can inhibit the oxidative stress response during the activation of T cells and maintain immune homeostasis [49]. Several recent studies have also confirmed the important role of glutamine/glutamate metabolism and/or the TCA cycle in allergy [21,25,45,46,50].

Several studies have highlighted the close association of histidine metabolism and arachidonic acid metabolism with anaphylaxis. Histamine, a histidine metabolite, is a well-known mediator of various pathophysiological events in anaphylaxis, but its short half-life limits its utility in diagnostic applications. Although the alterations in the metabolism of histidine and histamine were not observed, we identified up-regulation of their downstream metabolites imidazole acetic acid and methylimidazoleacetic acid. Moreover, imidazole acetic acid was identified as a potential biomarker of anaphylaxis by random forest feature screening and validation. However, the VIP values of both imidazole acetic acid and methylimidazoleacetic acid were not very high in plasma samples. It is probably because they were primarily metabolized through urine, and perhaps their changes could be more significant in urine. Unfortunately, the small volume of urine collected in anaphylaxis precluded further analysis in our study. Arachidonic acid (AA) is a precursor of many inflammatory mediators, including prostaglandins, leukotrienes, and thromboxanes, and is closely related to the occurrence of anaphylaxis. In the present study, the AA content was significantly up-regulated in FA, but no significant metabolic changes were observed in its downstream products. This likely resulted from the low content and short half-life of these downstream products.

The metabolic feature that distinguishes AMI is the disorder of linoleic acid metabolism. Linoleic acid, docosahexaenoic acid, and other unsaturated fatty acids help to reduce cholesterol and triglyceride, especially the concentration of LDL-c, and help to reduce the risk of coronary death [51]. Linoleic acid is a precursor of the AA biosynthesis. The metabolism of linoleic acid and AA metabolic abnormalities may lead to abnormal lipid metabolism. They can reduce heart contraction, decrease cardiac output, and even induce or aggravate myocardial infarction [52]. In the CAA group, the metabolism of linoleic acid to AA decreased, while its involvement in fatty acid degradation pathways increased (Figure 4). This may explain why anaphylaxis patients with severe cardiovascular symptoms usually exhibit fewer skin features.

Several metabolites in CAA were observed with opposite metabolic trends compared to FA. Some studies have found that a high-fat diet can exacerbate allergic reactions by affecting gut microbiota metabolism [32]. To investigate whether a high-fat diet and intestinal microbiota affect host metabolism, we conducted 16S rRNA analysis on fecal samples. The total number of ASVs. in FA and CAA was significantly higher than in AMI, and the increase in FA was more pronounced than in CAA. We found a reduction in the total number of ASVs. and the richness of gut microbiome in AMI after a high-fat diet, while not in CAA. This indicates that a high-fat diet can reduce the richness of the gut microbial community and, to a certain extent, can reverse the microbial changes caused by anaphylaxis.

The unique microbiota changes in the FA group were a significant increase in the abundance of *Lachnospiraceae_NK4A136_group* and a decrease in *Lactobacillus*. Du et al. [53] found a substantial increase in the relative abundance of *Lachnospiraceae_NK4A136_group* in subjects with wheat-dependent exertion-induced anaphylaxis (WDEIA), while the relative abundance of *Lactobacillus* was significantly decreased. Wang et al. [54] found that a β-lactoglobulin allergy increases the relative abundance of the Lachnospiraceae_NK4A136_group genus. These results are consistent with our findings. Some researchers have reported the anti-allergic effect of exogenous *Lactobacillus* supplementation [55,56,57]. *Lactobacillus* species showed a tendency to induce Th1 cytokines and inhibit Th2-biased responses in allergic reactions, which can regulate Th1/Th2 immune balance [58,59]. *Lactobacillus* may be a potential immunomodulatory agent for the treatment of FA [60,61]. However, after atherogenesis induced by high-fat feeding, the abundance of *Lactobacillus* in the CAA group was significantly increased. Similarly to our finding, Lee et al. [62] found that *Lactobacillus* exhibited a higher abundance at the genus level in mice fed a high-fat diet sensitized with Dermatophagoides pteronyssinus extract. They suggested that an increased abundance of *Lactobacillus* may lead to a declining exacerbation of obesity-related asthma.

Subsequently, a combination of three microbial genera (*g_Prevotellaceae_Ga6A1_group*, *g_UCG_008*, and *g_Eubacterium_hallii_group*) and three metabolites (tryptophan, trans-3-indole acrylic acid, and imidazole acetic acid) were selected from the candidate biomarkers based on the random forest classification algorithm for distinguishing anaphylaxis and non-anaphylaxis groups. Interestingly, all three genera were derived from candidate biomarkers in the AMI group. This implies uncertainty in microbial detection in anaphylactic patients. They are more susceptible to dietary patterns, especially changes in dietary structure, due to food allergies in some patients. When cardiovascular symptoms are present and anaphylaxis is suspected, detecting microbes that are highly expressed in AMI may be more reliable. In addition, a recent study by De Paepe et al. showed that gut microbiota changes precede allergic inflammation [63]. It further suggests that gut microbiota may be more suitable for preemptive prevention than post hoc FA diagnosis.

Compared with microbiota, metabolites are more robust for the diagnosis of anaphylaxis. All three metabolites can achieve high diagnostic performance in patients with allergies alone or with coronary atherosclerosis. In addition to the histamine metabolite imidazole acetic acid, tryptophan and trans-3-indole acrylic acid served as important biomarkers to distinguish anaphylaxis from non-anaphylaxis. They were down-regulated in both FA and CAA groups. Although these two metabolites were not listed as differential metabolites in FA, other tryptophan metabolites, including kynurenine, indole, and 3-indoxyl sulfate, were identified as differential metabolites. These tryptophan metabolites were similarly identified in CAA. The role of tryptophan and its metabolites in allergy has been widely reported. Consistent with our results, Zhang et al. [45] found that tryptophan and its metabolites were down-regulated in children with a milk allergy, and this was more evident in polysensitized than monosensitized children. Moreover, a significant down-regulation of tryptophan was detected in postmortem human blood samples collected from fatal anaphylaxis, and its diagnostic performance for the diagnosis of anaphylaxis and non-anaphylaxis death reached 87.1528, which was significantly higher than that of IgE and tryptase. It suggested the application prospect of tryptophan as a reliable biomarker for postmortem diagnosis of fatal anaphylaxis.

Wang et al. [64] found that 3-indole acrylic acid was significantly reduced in food-allergic mice and revealed that the dysregulation of indole metabolism was associated with a decrease in the abundance of beneficial bacteria, such as *Lactobacillus*. However, we did not find a regulatory effect of *Lactobacillus* on indole metabolites in the metabolite–microbe correlation network in FA. Instead, *Lactobacillus* was up-regulated and negatively correlated with indole metabolites in the CAA group. Moreover, the tryptophan metabolites of the indole pathway and kynurenine pathway were positively correlated with *Bacteroides* in the FA group but negatively correlated with *Bacteroides* and *Lactobacillus* in the CAA group. The phenylalanine metabolite phenylacetylglycine showed similar changes. The significant increase in *Lactobacillus* in CAA may be caused by the high-fat diet [65,66]. Chen et al. [67] confirmed that *Lactobacillus* intervention could increase the relative abundance of *Bacteroides* and down-regulate the levels of tryptophan, phenylalanine, arachidonic acid, and other metabolites to improve allergic reactions. Therefore, although many studies showed the promoting effect of a high-fat diet on allergies, the addition of a moderate amount of fat in the diet may have some protective effect on allergies by increasing the abundance of *Lactobacillus* [68,69].

Alternatively, we found a strong association of *Clostridium_sensu_stricto_1* with arginine in FA. This regulatory role appears to have not been reported. Several studies have found a protective effect of *Clostridium_sensu_stricto_1* against allergies in early life, but it gradually decreases over time in healthy children [27,70]. The association between *Clostridium_sensu_stricto_1* and short-chain fatty acids (SCFAs) in feces was highlighted [27], but the present study did not examine fecal metabolites, and no alteration of SCFAs in plasma metabolites in FA was discovered. Another interesting one in the correlation network of the CAA group is 2,3-dinor prostaglandin E1, which is related to *f_Prevotellaceae*, *f_Ruminococcaceae*, and *g_Christensenellaceae_R-7_group*. No biological activity of 2,3-dinor prostaglandin E1 has been reported, but it is derived from prostaglandin E1 metabolism. Prostaglandin E1, a metabolite of arachidonic acid, is a physiologically active substance with vasodilating and inhibiting platelet aggregation. It may be induced by the dual effects of atherosclerosis and anaphylaxis.

Our findings revealed the characteristics of plasma metabolites and gut microbiota in FA that are different from AMI and provided new insights for the diagnosis and treatment of anaphylaxis with isolated cardiovascular symptoms. A key strength of our study is the generation of isolated or comorbid rat models (FA, AMI, and CAA) to mimic patients with different conditions. However, some limitations must be addressed in future studies. Firstly, while effectively replicating core pathophysiological features, inherent species differences exist in coronary anatomy and immune regulation between rats and humans. The IgE paradox observed in comorbid CAA suggests divergent immune pathway activation requiring validation in humanized models or clinical cohorts. Moreover, although tryptophan depletion showed high discriminative power (AUC = 87.1528) in FA in postmortem human blood samples, we did not quantify the level of tryptophan. A large-scale, multicenter forensic autopsy study is needed to obtain more human samples for establishing the reference range of tryptophan in the future. Alternatively, urine would seem better as a noninvasive body fluid sample for metabolomics testing, but unfortunately, we collected almost no urine because of severe distributive shock due to anaphylaxis. In further study, prospective collection of antemortem urine in emergency settings may complement postmortem blood analysis. In addition, although it seems that tryptophan can still be detected at a long postmortem interval, its postmortem stability needs to be verified in future studies.

## 4. Materials and Methods

### 4.1. Animals

Forty-eight specific pathogen-free male Sprague–Dawley rats, aged 6~8 weeks, weighing (220 ± 20) g, were purchased from Beijing Vital River Laboratory Animal Technology Co., Ltd. (Beijing, China). All rats were kept under a constant temperature (24 ± 2 °C) and humidity (55 ± 5%) with a 12 h/12 h light/dark cycle. The rats were fed standard laboratory chow with water ad libitum and were acclimated to the room for 1 week. All animal experiments were approved by the Scientific Research Ethics Review Committee of Shanxi Medical University (No. 2019LL091, 7 March 2019).

### 4.2. Establishment of Animal Models

Forty-eight rats were randomly divided into four groups (n = 12), which were the fatal anaphylaxis group (FA), acute myocardial infarction (AMI) group, coronary atherosclerosis with anaphylaxis (CAA) group, and control (CON) group. For the FA group, ovalbumin (OVA, Grade V, A5503; Sigma-Aldrich, Saint Louis, MO, USA) was used to induce anaphylaxis according to the previous method [25,34]. The rats were sensitized by subcutaneously injecting 1 mL of OVA and 3.5 mg of Al(OH)_3_ (Sigma-Aldrich, Saint Louis, MO, USA)(dissolved in 1 mL 0.9% sterile saline) on Day 1, Day 4, and Day 14. Then, the sensitized rats were challenged intravenously with 10 mg of OVA(dissolved in 1 mL 0.9% sterile saline) on Day 21.

For the AMI group, the rats were gavaged with high-fat emulsion combined with an intraperitoneal injection of vitamin D3 (Sigma-Aldrich, Saint Louis, MO, USA) to induce coronary atherosclerosis [71]. A total of 25 mL of tween-80 (MeilunBio, Dalian, China), 5 g of cholesterol (Sangon, Shanhai, China), 5 g of white sugar(SantaiBio, Changzhou, China), 5 g of egg yolk powder (Yuanye, Shanghai, China), and 1 g of propylthiouracil (Solarbio, Beijing, China) were added into 25 g of melted lard (Yuanye, Shanghai, China) to make the oil phase mixture. In total, 2 g of sodium deoxycholate (MeilunBio, Dalian, China) was added into 50 mL of distilled water to make the aqueous phase mixture. The oil and aqueous phases were mixed evenly to make the high-fat emulsion. All rats were gavaged with the high-fat emulsion for 12 weeks (2 mL twice daily). At the beginning of the experiment, the rats were intraperitoneally injected with 600,000 IU/kg of vitamin D3, and another 100,000 IU/kg of vitamin D3 was intraperitoneally injected at Week 3, Week 6, and Week 9, respectively, for a total of 900,000 IU/kg. After 12 weeks of gavage, total cholesterol (TC), triglyceride (TG), low-density lipoprotein (LDL), and high-density lipoprotein (HDL) in the rat serum were measured via an automatic biochemical analyzer (TBA-FX8, Toshiba, Tokyo, Japan). The atherosclerosis index (AI) = (TC-HDL)/HDL was calculated, and AI ≥ 3.8 was regarded as successful modeling. After confirmation of coronary atherosclerosis in rats, the rats were anesthetized with isoflurane (RWD, Shenzhen, China) (2~3%). The rats were intubated by oral cannula and connected to a ventilator with a ventilation rate of 75 breaths/min, the tidal volume was 0.8~1 mL, and the respiratory ratio was 2:1. The limbs were connected to a BL-420F/A biofunctional system (Techman, Chengdu, China) to record the preoperative electrocardiogram. The skin of the third to fourth intercostal space with the strongest heart pulse was cut through, and the muscular layer tissue was bluntly separated. The pericardium was stripped to expose the heart. Finally, the left anterior descending coronary artery (LAD) with a small amount of myocardial tissue was ligated at 3~5 mm below the left atrial appendage to induce acute myocardial infarction [72]. The postoperative electrocardiogram was recorded. For the CAA group, the rat model of coronary atherosclerosis was established by the same method described above, then sensitized and challenged by the same method as the FA group. The rats of the control group were sacrificed without any treatment.

### 4.3. Sample Collection and Preparation

After model preparation, fecal samples were collected using a sterile centrifuge tube in a metabolic cage. The blood was collected by cardiac puncture at the end of the experiment. The collected blood was divided into two parts. In total, 2 mL of blood was left at 4 °C overnight and centrifuged at 10,000 rpm for 10 min at 4 °C to separate serum. The other 2 mL of blood was transferred into heparin sodium tubes and centrifuged at 10,000 rpm for 10 min at 4 °C to separate plasma. All samples were stored at −80 °C until further use. The organs of the rats were collected and fixed in 10% neutral formaldehyde.

### 4.4. Serum IgE and Tryptase Measurement and Tissue Staining

The serum IgE and tryptase were quantified using commercially available rat ELISA kits (AD1720Ra, Andy Gene, Beijing, China; SEB070Ra, Cloud-Clone, Wuhan, China) following the manufacturer’s instructions.

Three heart tissues from each group were dehydrated with 30% sucrose. The hearts were cut from the coronal view into five equal portions and embedded with an optimal cutting temperature compound (Sakura Finetek USA, Torrance, CA, USA) before cutting into 10 μm sections on the aortic arch, aortic valve, and coronary artery plane. The sections were stained with oil Red O (Sigma-Aldrich, Saint Louis, MO, USA) to evaluate the development of atherosclerosis.

The remaining tissues were embedded in paraffin and then cut into 4 μm sections for hematoxylin and eosin (HE) staining. Immunohistochemical (IHC) staining of lung tissues was performed with rat tryptase antibody (A14471, dilution 1:100; ABclonal Biotechnology, Wuhan, China).

### 4.5. Plasma Metabolomics Analysis

The plasma samples were thawed at 4 °C, and 100 μL was transferred into a 1.5 mL microcentrifuge tube. In total, 200 μL of 0.1% acetonitrile formate (Sigma-Aldrich, Saint Louis, MO, USA) was added and vortexed for 3min, then centrifuged at 13,000 rpm for 15 min at 4 °C. The prepared supernatant samples were taken for analysis. Quality control (QC) samples were prepared by pooling equal aliquots (10 μL) from each sample.

Metabolomics analysis was performed using the UltiMate 3000 ultra-performance liquid chromatography–tandem quadrupole electrostatic field orbital trap mass spectrometry (UHPLC-Q Exactive Orbitrap-MS, Thermo Fisher Scientific, Waltham, MA, USA). All tested plasma samples were randomly ordered for analysis to reduce the possible instrument system error during the whole analysis process. Before formal analysis, the QC samples were analyzed continuously five times to ensure reproducibility. To evaluate the precision of the LC-MS method across different batches, a blank and a QC sample were analyzed after every 10 experimental samples.

The chromatographic separation was performed using a Waters ACQUITYUPLC HSS T3 chromatographic column (2.1 mm × 100 mm, 1.7 μm; Waters, Milford, MA, USA). The mobile phase comprised 0.1% formic acid in ultrapure water (solvent A) and acetonitrile (solvent B). The gradient elution program started with 98% solvent A in the first 2 min, then decreased to 65% in the next minute. Over the following 15 min, the percentage of solvent A decreased to 30% and remained at 30% for 1 min. It then further decreased to 2% within the following 10 min and maintained for 2 min. From 31 to 33 min, the percentage of solvent A re-equilibrated back to 98% and remained for the last 2 min. The flow rate was consistently maintained at 0.2 mL/min, with the column temperature set at 40 °C and an injection volume of 5 μL.

The mass spectrometry detection employed an electrospray ionization (ESI) source in both positive and negative ion modes. Full Scan /dd-MS2 mode was utilized to obtain mass spectra, covering a mass range from 80 to 1200 *m*/*z*. The resolution was set to MS Full Scan 35,000 FWHM and MS /MS 17,500 FWHM, and normalized collision energy was set to 12.5 eV, 25 eV, and 37.5 eV. The voltage of the positive ion mode was 3.5 kV, while the negative ion mode was 2.5 kV. The temperature of the heater was 300 °C, and the temperature of the capillary was 320 °C. The sheath gas flow rate was 35 arb, and the auxiliary gas flow rate was 10 arb.

The raw metabolomics data were processed using Compound Discoverer software (version 3.1, Thermo Fisher Scientific, Waltham, MA, USA) to convert the format. After normalization, the data were performed multivariate statistical analysis using SIMCA-P software (version 14.1, Umetrics, Umea, Sweden). The principal component analysis (PCA), partial least squares-discriminant analysis (PLS-DA), and orthogonal partial least squares discrimination analysis (OPLS-DA) were performed to analyze the metabolic differences between the control and model groups. The differential metabolites were identified using variable importance in projection (VIP) > 1 and *p* < 0.05 as criteria. The MetaboAnalyst 5.0 (http://www.metaboanalyst.ca/, accessed on 28 February 2024) was used to identify metabolic pathways (MetPA).

### 4.6. 16S rRNA Sequencing

Microbial genomic DNA was extracted from thawed fecal samples using the TIANamp Stool DNA Kit (TIANGEN, Beijing, China) following the manufacturer’s protocol. The V3–V4 region of bacterial 16S rRNA was amplified by PCR using universal primers 341F (5′-ACTCCTACGGGAGGCAGCAG-3′) and 806R (5′-GGACTACHVGGGTWTCTAAT-3′). Amplicon products were purified using a Gel Extraction Kit (Qiagen, Dusseldorf, Germany) and quantified with a Qubit 2.0 Fluorometer (Thermo Fisher Scientific, Waltham, MA, USA). Sequencing libraries were prepared following the TruSeq^®^ DNA PCR-Free Sample Preparation Kit (Illumina, San Diego, CA, USA) and then sequenced on an Illumina novaseq6000 platform (Illumina, San Diego, CA, USA).

Paired-end reads were assigned to samples based on their unique barcode. The raw reads were subjected to debarcode and deadapter, quality filtering, and the removal of chimeras to obtain clean reads according to the QIIME2 quality control process. The Divisive Amplicon Denoising Algorithm (DADA2) in QIIME2 was used for denoising to obtain amplicon sequence variants (ASVs.). Instead of operational taxonomic units (OUTs) clustering by 97% similarity, it performs only dereplication or clustering by 100% similarity. DADA2 is more sensitive and specific than the traditional OTU method, which improves the accuracy, comprehensiveness, and reproducibility of genomic data analysis. The Silva138 database and a pre-trained Naive Bayes classifier based on the classify-sklearn algorithm in the QIIME2 were used for each ASV to annotate taxonomic information.

The alpha diversity between groups was estimated using Chao1, ACE, Shannon, and Simpson diversity indexes. The differences between groups were compared using the Wilcoxon test (pairwise comparison) and Kruskal–Wallis test (multiple comparison). The beta diversity was estimated using Principal Co-ordinates Analysis (PCoA) based on the Bray–Curtis distance. The analysis of similarities (Anosim) was used to compare group differences in microbial communities. The linear discriminative analysis effect size (LEfSe) was performed to identify differential genera between groups.

### 4.7. Random Forest Classification Model

The receiver-operating characteristic (ROC) curve was constructed for the identified differential metabolites and microbial genera, and the area under the curve (AUC) was calculated. Metabolites and microbial genera with AUC ≥ 0.7 were selected as candidate biomarkers for subsequent analysis. The random forest was used to evaluate the importance of features for 2/3 of the samples by R (version 4.0.3). Then, the most important features (a set of metabolites and microbial genera combinations) were used to construct the classification model, which was verified on the remaining 1/3 of the samples.

### 4.8. Correlation Analysis

The relationships among metabolites and microbiomes were investigated using Spearman correlation analysis by R (version 4.0.3). The network between altered genera and metabolites was visualized by Cytoscape (version 3.8.2) with significant correlation coefficients |r| ≥ 0.5 and *p* < 0.05.

### 4.9. Forensic Samples

The human blood samples were collected in the process of forensic autopsy from the Forensic Science Center of Shanxi Medical University. The subjects included death from fatal anaphylaxis, acute myocardial infarction, and other causes (n = 12). The causes of death were determined by three or more forensic pathologists, and informed consent was obtained from the family of the deceased and the forensic institution. The serum IgE and tryptase were quantified using commercially available ELISA kits (SEA545Hu and SEB070Hu, Cloud-Clone, Wuhan, China) following the manufacturer’s instructions. Metabolomics was performed on human blood samples as described in Section 4.5. The experiments were approved by the Scientific Research Ethics Review Committee of Shanxi Medical University (No. 2022GLL093, 3 March 2022).

### 4.10. Statistics Analysis

The measurement data were analyzed with GraphPad Prism 8.0 software, and all results were presented as mean ± standard deviation (SD). When data conformed to a normal distribution, the *t*-test was used for pairwise comparison, and the one-way analysis of variance (ANOVA) was used for multiple comparisons followed by the Holm’s Stepdown Bonferroni procedure for adjusted *p*-values. For data that did not conform to a normal distribution, the Wilcoxon rank sum test was used for pairwise comparison, the Kruskal–Wallis test was applied to multiple groups, and the Bonferroni method was used for correction between groups.

## 5. Conclusions

In conclusion, we highlighted the potential of plasma metabolites as biomarkers to determine whether FA has occurred. Tryptophan (AUC = 87.1528) may be a reliable postmortem biomarker of FA, which was significantly better than that of serum IgE (AUC = 44.0972) and serum tryptase (AUC = 67.7083). Fecal microbes may be more suitable for the ex-ante prediction of FA. Different dietary structures can change the microbiota of FA. The addition of appropriate high-quality fat to the diet may be beneficial in increasing the abundance of Lactobacillus and reducing the severity of allergic reactions.

## Figures and Tables

**Figure 1 ijms-26-06292-f001:**
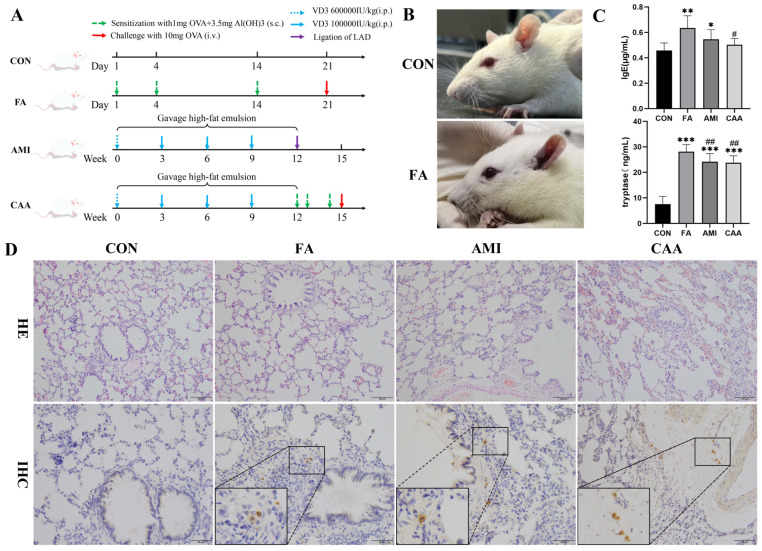
The characteristics of fatal anaphylaxis: (**A**) Schematic diagram of rat model preparation. (**B**) The cyanotic mouth, nose, and auricle of fatal anaphylaxis rats after the OVA challenge. (**C**) Serum IgE and tryptase of rats. * *p* < 0.05 vs. control, ** *p* < 0.01 vs. control, *** *p* < 0.001 vs. control, # *p* < 0.05 vs. FA, ## *p* < 0.01 vs. FA. (**D**) The representative sections of a lung stained with HE (200×) and tryptase IHC (400×). LAD: left anterior descending coronary artery.

**Figure 2 ijms-26-06292-f002:**
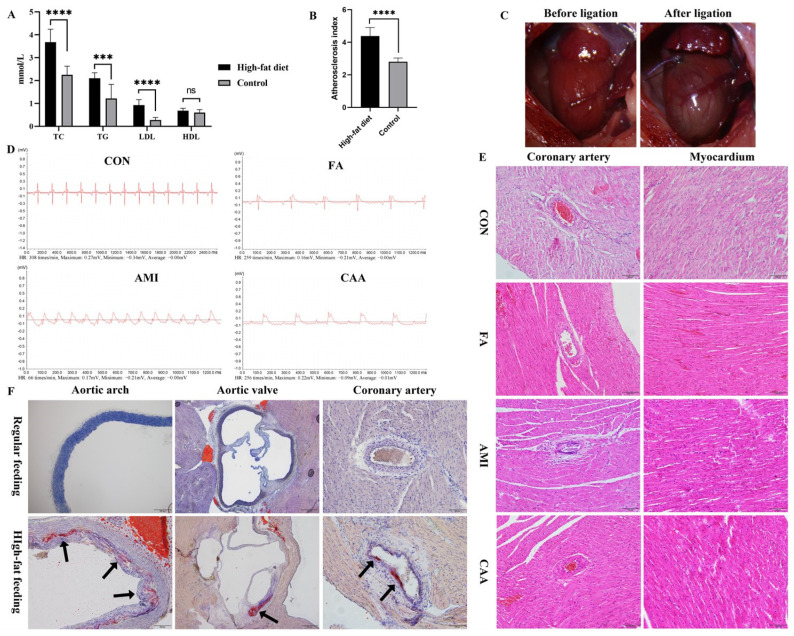
The characteristics of high-fat-diet rats and after coronary artery ligation: (**A**) The serum lipid levels in high-fat diet rats, including TC, TG, LDL, and HDL. (**B**) The serum atherosclerosis index in high-fat-diet rats. (**C**) Myocardial ischemia and pallor in the left ventricle after ligation of LAD. (**D**) The representative electrocardiograms of each group. (**E**) The representative sections of the coronary artery and myocardium stained with HE (200×). (**F**) The representative sections of the aortic valve (50×), aortic arch (100×), and coronary artery (200×) stained with oil red O. The arrows show the atherosclerotic plaques stained bright red with oil red O. “ns” represents no significance, *** *p* < 0.001, **** *p* < 0.0001. TC, total cholesterol; TG, triglyceride; LDL, low-density lipoprotein; HDL, high-density lipoprotein; AI, atherosclerosis index.

**Figure 3 ijms-26-06292-f003:**
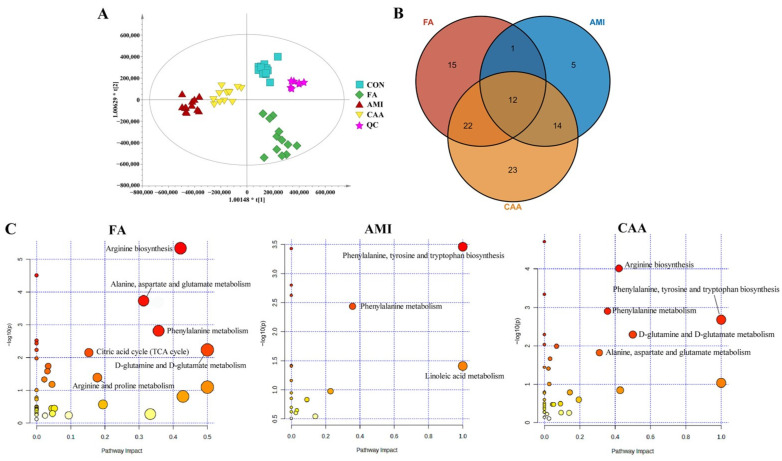
The analysis of metabolomics. (**A**) OPLS-DA scatter plot of overall samples. (**B**) The Venn diagram of differential metabolites. (**C**) The MetPA diagrams.

**Figure 4 ijms-26-06292-f004:**
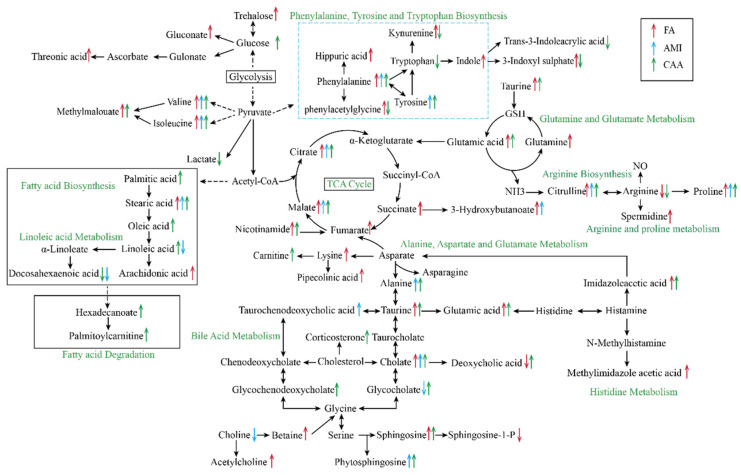
The altered metabolites and perturbed metabolic pathways. The altered metabolites were labeled with up-regulated (↑) and down-regulated (↓). The red, blue, and green arrows indicate changes in the FA, AMI, and CAA groups, respectively. The green text indicates perturbed metabolic pathways.

**Figure 5 ijms-26-06292-f005:**
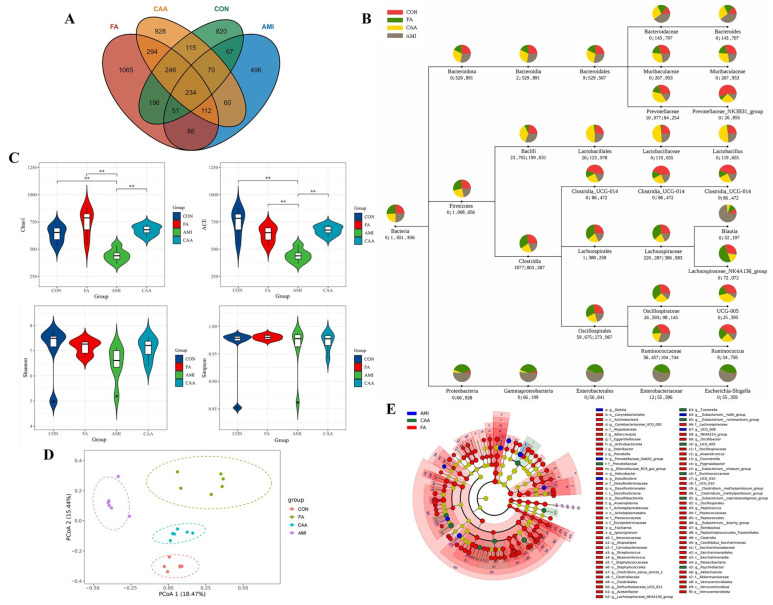
The altered gut microbiota. (**A**) The Venn diagram of ASVs. (**B**) The taxonomy analysis tree. The colored sectors represent different groups, and the size of each sector indicates the proportional abundance at the taxonomic level. The first number under the circle denotes the count of sequences that only align with that specific classification (not to a lower classification rank), and the second number indicates the total count of aligned sequences. (**C**) The violin diagrams of alpha diversity. Each box plot showed the minimum, first quartile, median, third quartile, and maximum value of the sample index in the group. ** *p* < 0.01. (**D**) The PCoA analysis based on Bray–Curtis distances indicated the beta diversity. (**E**) The cladogram. The bar chart shows the classification of species with significant effects in different groups. The colored nodes in the branches represent the microbiota that plays an important role in each group, and the yellow nodes represent the adiaphorous microbiota.

**Figure 6 ijms-26-06292-f006:**
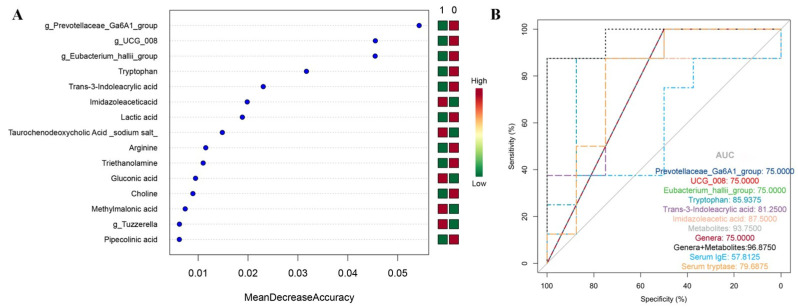
The random forest classification model. (**A**) The feature importance ranking: 1, anaphylaxis (including the FA and CAA groups); 0, non-anaphylaxis (including the CON and AMI groups). (**B**) Validation ROC curves of the random forest classification model consisting of the most important three microbial genera and three metabolites.

**Figure 7 ijms-26-06292-f007:**
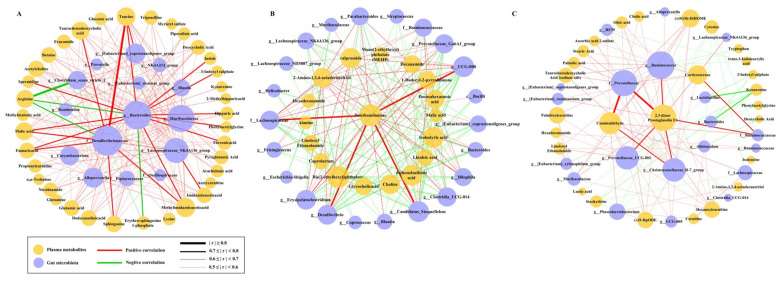
Integrated networks of metabolites and gut microbiota. The size of the circle in the network represents the degree of connectivity, while the thickness of the line represents the strength of the correlation. (**A**) Integrated networks of the FA group. (**B**) Integrated networks of the AMI group. (**C**) Integrated networks of the CAA group.

**Figure 8 ijms-26-06292-f008:**
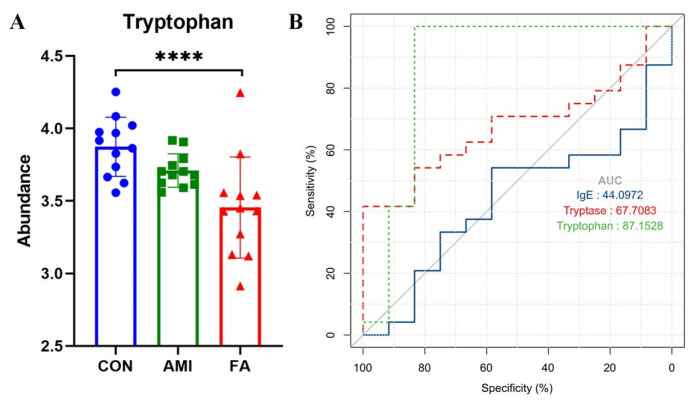
(**A**) The abundance of tryptophan in different causes of death. **** *p* < 0.0001. (**B**) The ROC curve showed the diagnostic performance of tryptophan in distinguishing FA from non-FA better than serum IgE and tryptase.

**Table 1 ijms-26-06292-t001:** The characteristics of forensic samples.

Group	Cause of Death/Allergen	N	Male/Female	Age (years)	PMI (h)	IgE (ng/mL)	Tryptase (μg/mL)
CON	Amniotic fluid embolism	1	10/2	31.50 ± 19.62	203.50 ± 127.52	375.76 ± 351.61	30.87 ± 12.25
	Death from cold	1					
	Fat embolism	1					
	Sepsis	1					
	Hemorrhagic shock	4					
	Cerebral hemorrhage	2					
	Somedon poisoning	1					
AMI	Coronary atherosclerotic heart disease	11	11/1	51.83 ± 15.20	331.50 ± 221.5	102.77 ± 191.63	33.08 ± 14.53
	Muscular bridging of the right coronary artery	1					
FA	Cefotaxime	2	9/3	42.00 ± 17.63	203.00 ± 192.49	363.71 ± 517.35	27.70 ± 5.83
	Cefoperazone Sodium and Sulbactam Sodium	1					
	Penicillin	1					
	Ampicillin	2					
	Levofloxacin	2					
	Acetaminophen	1					
	Clindamycin	1					
	Azithromycin	1					
	Ribavirin	1					

## Data Availability

All data used for the analyses in this report are available from the corresponding author upon reasonable request.

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
