# Peer review of "Integrated Metabolomic and Gut Microbiome Profiles Reveal Postmortem Biomarkers of Fatal Anaphylaxis"

_ijms, 2025, doi:10.3390/ijms26136292_

Round 1

Reviewer 1 Report

Comments and Suggestions for Authors

The authors present an interesting study in which they analyze metabolomic and gut microbiome profiles in rat models of fatal anaphylaxis (FA), acute myocardial infarction (AMI), and coronary atherosclerosis with anaphylaxis (CAA), aiming to identify novel biomarker(s) of fatal anaphylaxis. Among the candidate biomarkers identified, three were further validated in a small number of human forensic samples of fatal anaphylaxis, with tryptophan emerging as a promising postmortem biomarker.

The manuscript may benefit from addressing the following major points, which could potentially strengthen its scientific merit:

  1. One of the main limitations is the reliance on the current rat models of FA, AMI, and CAA. The authors must clarify the pathophysiological rationale behind these models and provide stronger evidence of their relevance to the human conditions. For instance, how do the authors explain elevated IgE levels in the AMI model, but not in CAA? Additionally, what do the brown tryptase-positive granules observed in the lungs of the AMI group signify? What is the proposed role of mast cells in the context of AMI?
  2. The authors should include a comparative analysis of established postmortem serum markers of anaphylaxis, such as tryptase and chymase. Tryptase, in particular, is a well-recognized biomarker in forensic investigations of fatal anaphylaxis (e.g., PMID: 29331682) and should be discussed and, measured in the current study.
  3. The manuscript would benefit from a more comprehensive discussion of known and published biomarkers of anaphylaxis. How do the newly proposed markers compare with established ones in terms of sensitivity, specificity, and postmortem stability?
  4. The authors state in the Introduction that elevated IgE is associated with fatal anaphylaxis. Please provide an appropriate reference to support this claim.

Author Response

Thank you very much for taking the time to review this manuscript. Please find the detailed responses below and the corresponding revisions in the resubmitted manuscript.

Comments 1: One of the main limitations is the reliance on the current rat models of FA, AMI, and CAA. The authors must clarify the pathophysiological rationale behind these models and provide stronger evidence of their relevance to the human conditions. For instance, how do the authors explain elevated IgE levels in the AMI model, but not in CAA? Additionally, what do the brown tryptase-positive granules observed in the lungs of the AMI group signify? What is the proposed role of mast cells in the context of AMI?

Response 1: (1) The pathophysiological rationale of rat models: OVA-induced anaphylaxis is the most commonly used animal model to study anaphylaxis. Its pathophysiological rationale is based on IgE-mediated type I hypersensitivity, and the core mechanism is highly similar to that of human anaphylaxis. The pathophysiological mechanism of atherosclerosis induced by high-fat feeding in rats is discussed in the third paragraph of the discussion section of the manuscript (lines 355-362). The ligation of coronary arteries based on atherosclerosis more closely resembles the pathological process in humans and helps to reproduce the clinicopathological features of acute myocardial infarction based on atherosclerosis. The CAA model integrates the above two interventions to simulate the complex state of patients with comorbidities, such as those with coronary artery disease who develop anaphylaxis.

(2) The explanation of the difference in IgE levels between AMI and CAA groups: The evaluated IgE level in the AMI group is consistent with clinical studies, and IgE can promote coronary plaque instability by activating mast cells to release inflammatory factors. The absence of elevated IgE levels in the CAA group may be due to the preferential activation of the IgG/FcγR pathway (non-IgE-dependent pathway) by the atherosclerotic microenvironment, and the alternative pathway mediates allergic reactions. In addition, atherosclerosis induced by a high-fat diet belongs to the Th1-type immune response, while anaphylaxis is a Th2-type immune response. The Th1/Th2 balance is crucial for maintaining the normal function of the immune system and preventing the occurrence of diseases. In the presence of atherosclerosis, the cytokine microenvironment shifts the Th1/Th2 immune balance toward Th1-type. When anaphylaxis occurs, a Th1-type immune response may inhibit the proliferation and differentiation of B cells and the production of IgE. Jaramillo et al. (reference 18) also found lower allergen-specific IgE (sIgE) in patients with a history of myocardial infarction and pointed out that the mutual antagonism between Th1/Th2 may make atopy an independent protective factor for MI, which is consistent with our results. The relevant content is supplemented in the discussion of the CAA model in the fourth paragraph of the revised manuscript discussion section (lines 368-390).

(3) The significance of brown tryptase-positive granules observed in the lungs of AMI patients and the role of mast cells in acute myocardial infarction: Mast cells normally populate the myocardium and adventitia of coronary arteries. Mast cells accumulate and become activated in areas of atherosclerotic plaque erosion or rupture, increasing with the clinical severity of the coronary syndrome. Myocardial ischemia activates mast cells to release tryptase, which promotes increased vascular permeability, matrix remodeling, and inflammation. The positive expression of tryptase in the lung tissue indicates systemic mast cell activation, which is consistent with the systemic inflammatory state of AMI. This mast cell activation in the non-allergic state suggests that the postmortem diagnosis of anaphylaxis based on tryptase may be false positive.

Comments 2: The authors should include a comparative analysis of established postmortem serum markers of anaphylaxis, such as tryptase and chymase. Tryptase, in particular, is a well-recognized biomarker in forensic investigations of fatal anaphylaxis (e.g., PMID: 29331682) and should be discussed and measured in the current study.

Response 2: Serum tryptase is indeed a recognized biomarker in forensic investigations of fatal anaphylaxis. Only immunohistochemical staining for tryptase was performed in our previous manuscript because we considered that serum tryptase in cadaver samples is susceptible to factors such as sampling site and hemolysis. In the revised manuscript, we have added the results of serum tryptase assays for rat and human samples (Figure 1, Table 1).

Comments 3: The manuscript would benefit from a more comprehensive discussion of known and published biomarkers of anaphylaxis. How do the newly proposed markers compare with established ones in terms of sensitivity, specificity, and postmortem stability?

Response 3: In Section 2.6 of the manuscript, we supplemented the serum IgE and tryptase levels of human samples (Table 1), and evaluated the diagnostic performance of tryptophan and these two commonly used biomarkers of anaphylaxis by ROC curves (Figure 8B). The AUC value of tryptophan was significantly higher than that of IgE and tryptase, indicating that tryptophan had better sensitivity and specificity. The consistency of changes in tryptophan levels in human samples collected some time after death with those in samples collected immediately after death in rats indirectly proves its postmortem stability, but it still needs to be verified in future studies.

Comments 4: The authors state in the Introduction that elevated IgE is associated with fatal anaphylaxis. Please provide an appropriate reference to support this claim.

Response 4: Thank you for pointing out the omissions in our writing. Although most studies consider tryptase to be a good biomarker for postmortem diagnosis of anaphylaxis, some studies have highlighted the significance of IgE detection in serum and tissues for postmortem diagnosis of anaphylaxis. The relevant literature is supplemented in the third paragraph of the introduction (References 13-15).

Reviewer 2 Report

Comments and Suggestions for Authors

The topic of the research is potentially important; however, it seems that the paper is overly complicated with many unnecessary experiments.  Table 1 shows a good collection of well-characterizd forensic specimens, which should have been sufficient to develop a biomarker distinguishing FA from other conditions.  What was the point of killing a whole bunch of rats to determine that tryptophan works for distinguishing the human specimens?  Also, what was the significance of doing the 16S rRNA analysis of the rat fecal microbiome?  None of this information seemed to be relevant for the human specimens.  If it was up to me, I would only focus on human specimens, analyze the metabolomics, do the random forest model and figure out what metabolite is a good biomarker to discriminate FA from other conditions.  Spare the rats...

Minor comments: there are too many abbreviations.  Make a short table of abbreviations at the beginning of the paper.

Author Response

Thank you very much for taking the time to review this manuscript and for your affirmation of our research topic. Please find the detailed responses below and the corresponding revisions in the resubmitted manuscript.

Comments 1: The topic of the research is potentially important; however, it seems that the paper is overly complicated with many unnecessary experiments.  Table 1 shows a good collection of well-characterizd forensic specimens, which should have been sufficient to develop a biomarker distinguishing FA from other conditions. What was the point of killing a whole bunch of rats to determine that tryptophan works for distinguishing the human specimens? Also, what was the significance of doing the 16S rRNA analysis of the rat fecal microbiome? None of this information seemed to be relevant for the human specimens. If it was up to me, I would only focus on human specimens, analyze the metabolomics, do the random forest model and figure out what metabolite is a good biomarker to discriminate FA from other conditions. Spare the rats...

Response 1: (1) We understand your concerns about the complexity of the experimental design and the ethical issues surrounding the use of animals. We strongly agree with your view to focus only on human samples and perform metabolomics analysis to identify metabolic biomarkers to distinguish fatal anaphylaxis (FA). Indeed, this is how our study was initially conducted. However, since drugs are the primary cause of FA in China, FA typically occurs during hospitalization. This means that patients usually receive timely drug rescue and first-line treatment following the onset of FA. The use of allergenic and rescue drugs complicates the differential metabolites in the plasma of FA patients compared to the control group. The differential metabolites of FA patients identified by the random forest model were mainly these drugs and their secondary metabolites. On the other hand, the postmortem interval of our human samples was prolonged, and some key metabolites may degrade due to decomposition, which could diminish the accuracy of the results. Therefore, we aimed to control the variables precisely by creating a rat model to eliminate confounding factors and the effects of postmortem changes. The results of our animal model confirmed that the disorder of tryptophan metabolism was a direct consequence of anaphylaxis itself rather than a postmortem change or confounding factor. And the rat model mimicked the pathological features of FA and acute myocardial infarction (AMI) in humans, demonstrating that tryptophan can distinguish FA in coexisting disease states. The reliability of tryptophan was also verified in human samples (AUC=87.15, Figure 8). In addition, the consistency of tryptophan metabolism in rat blood immediately after death and human samples collected some time after death proves the stability of tryptophan as a biomarker for forensic identification of FA. While we regret the sacrifice of animal lives, their contribution was ethically justified and minimized to resolve a critical forensic dilemma of FA diagnosis. All animal experiments were approved by the Scientific Research Ethics Review Committee of Shanxi Medical University (No. 2019LL091), and we fully followed the 3R principles and ARRIVE 2.0 guidelines in the experiment.

(2) In response to your question about the significance of 16S rRNA analysis of the rat fecal microbiota, we provided in the introduction that gut microbes affect the immune status of the body by affecting host metabolites (lines 89~103). The different diet structure may have an impact on the body's gut microbes, which may cause changes in the composition of metabolites. Since a high-fat diet is required for the construction of AMI and CAA models in rats, it is necessary to explore the correlation between gut microbiota and metabolites. Through the metabolites-microbiota correlation network (Figure 7), we found a correlation between tryptophan metabolism and Bacteroides and Lactobacillus. A high-fat diet may reduce the severity of anaphylaxis by increasing Lactobacillus abundance, which provides a basis for future prevention strategies based on microbiota intervention. In addition, noninvasive biomarkers of gut microbes as allergic reaction seem to be very promising, but few studies have been reported in the forensic diagnosis of FA. We aimed to identify potential biomarkers of FA by 16S rRNA sequencing, but the microbial markers identified by the random forest model were not FA but differential microorganisms of AMI. The ROC curve also confirmed that the diagnostic efficacy of microorganisms alone (AUC=75.0) was lower than that of serum metabolites (AUC=93.75) (Figure 6). This suggested that microorganisms are not good biomarkers for the diagnosis of FA, so we did not further investigate them in human samples.

Comments 2: There are too many abbreviations. Make a short table of abbreviations at the beginning of the paper.

Response 2: Thanks for your suggestion, we added a table of abbreviations below the abstract (line 30) and removed some unnecessary abbreviations in the manuscript. The modification was made in a revision mode and can be found in the resubmitted manuscript.

Round 2

Reviewer 1 Report

Comments and Suggestions for Authors

While the authors have improved the manuscript, my primary concern remains the translational value and validity of the animal models used. It remains unclear to what extent the employed rat models accurately recapitulate the three complex human pathologies studied, particularly in the context of FA and overlapping cardiovascular conditions.

Author Response

Comment: While the authors have improved the manuscript, my primary concern remains the translational value and validity of the animal models used. It remains unclear to what extent the employed rat models accurately recapitulate the three complex human pathologies studied, particularly in the context of FA and overlapping cardiovascular conditions.

Response: We are grateful to the reviewers for rereviewing our manuscript. We deeply understand the continued focus on the translational value and validity of the animal models used. This is indeed a key aspect of this study, and we elaborate in more depth below to provide additional evidence to clarify the scientific basis of the models and their value in the context of this study. In addition, a description of the limitations of animal models has been added to the discussion section of the manuscript (lines 559-562), which can be found in the resubmitted manuscript.

(1) Fatal anaphylaxis (FA) model: The classical rat FA model of ovalbumin sensitization/challenge was chosen for its wide acceptability and ability to recapitulate key characteristics of human IgE-mediated anaphylaxis (references 25 and 34). The characteristics of FA include rapidly occurring hypotension/shock, airway spasm, increased vascular permeability, mast-cell degranulation (elevated serum tryptase levels, degranulation of lung mast cells), eosinophilia, and ultimately death from cardiovascular failure. The anaphylactic symptoms, mortality, elevated serum IgE and tryptase, and histopathological changes (such as pulmonary congestion and edema, bronchospasm, increased eosinophils, and mast cell tryptase expression) of OVA-induced FA rats were shown in the results section (Section 2.1, Figure 1B-D). We replaced the HE section of the FA group in Figure 1D to show more pronounced bronchoconstriction, to characterize the severe respiratory symptoms that occurred after OVA challenge. Unfortunately, due to laboratory constraints, we were not able to monitor the blood pressure of the rats after OVA challenge, but the electrocardiogram (Figure 2D) showed a significant reduction in heart rate after OVA challenge, which is consistent with the results of previous studies (reference 34). These results confirmed the validity of the OVA-induced FA model. Importantly, the key metabolite changes observed by our model are consistent with some of the results reported in human anaphylaxis/allergic disease studies, providing indirect support for the model's relevance.

(2) Acute myocardial infarction (AMI) model: The model of feeding with a high-fat diet and injecting VD3 has been widely used in the research related to atherosclerosis in rats (e.g., PMID: 36232451; PMID: 38845851). However, a high-fat diet has poor palatability, which can easily lead to large intra-group variability due to individual food intake. Therefore, we modified the high-fat diet to a high-fat emulsion by gavage according to reference 71 (new reference) to ensure the homogeneity of the model. Permanent ligation of the left anterior descending coronary artery is one of the most established and widely validated models for establishing experimental AMI (new reference 72), reliably inducing myocardial ischemia, necrosis, characteristic electrocardiographic changes, elevated cardiac biomarkers, and predictable worsening of cardiac function and mortality. The core links of these pathophysiological processes (myocardial ischemia, energy metabolism crisis, inflammatory response) are highly similar to those of human AMI. Atherosclerotic changes in the rat arterial endothelium were confirmed by measurement of lipid levels to calculate the atherosclerosis index (Figure 2A-B), HE staining, and oil red O staining (Figure 2E-F). And the occurrence of acute myocardial infarction was confirmed by sustained ST-segment elevation on electrocardiography (Figure 2D) and pathological changes in myocardial tissue (Figure 2C, E). These verification results of successful model establishment are provided in Section 2.1. The reviewer may be concerned about the validity of the AMI model because of the previously mentioned issues of elevated serum IgE and the positive tryptase (which were addressed in detail in the first revision). However, some previous studies have demonstrated the elevation of serum IgE and local mast cell activation in human AMI (e.g., PMID: 30599216; PMID: 3286037; PMID: 7648650; PMID: 40103802). We believe that the elevated IgE and tryptase levels in the AMI rat model are not evidence of the model's ineffectiveness but rather a manifestation of its complexity, consistent with what has been observed in some human AMI studies.

(3) Coronary atherosclerosis with anaphylaxis (CAA) model: This model is designed to simulate the complex clinical scenario of anaphylaxis superimposed on pre-existing cardiovascular diseases, such as atherosclerosis, and is the superposition of the previous two models. Although there is a lack of studies on the comorbidity of atherosclerosis and fatal anaphylaxis, many researchers have investigated the pathological changes when obesity induced by high-fat diet feeding coexists with OVA-induced asthma, allergic rhinitis, or food allergy (e.g., PMID: 23005257; PMID: 40252413; PMID: 23840965). In this study, the basis of atherogenesis induced by high-fat feeding was performed at the same time as in the AMI group, and stable atherosclerosis was confirmed by lipid measurements and increased atherosclerosis index, ST-segment elevation, and decreased heart rate on electrocardiography, oil red O and HE staining results (Figure 2). This was followed by OVA sensitization and challenge, which resulted in the development of a fatal anaphylactic phenotype (including the development of allergic symptoms, a decrease in heart rate, bronchoconstriction, increased eosinophils in lung tissue, and increased tryptase expression in serum and tissue) (Figure 1). This model presents an overlay of partial features of FA and AMI, but also shows unique patterns, such as differences in metabolic pathways described in the results, supporting the potential of this model for studying such interactions. The absence of elevated IgE in serum may have contributed to the reviewer's concerns about the validity and translational value of the CAA model. The mechanism of OVA-induced anaphylaxis should be IgE-mediated in theory, but the occurrence of anaphylaxis also involves a variety of other mechanisms, including IgG, complement, or Mas-related G protein-coupled receptors. In the complex pathological context of atherosclerosis, mast cell activation may shift to a non-IgE-mediated pathway. Since mast-cell tryptase release of CAA (Figure 1C, D) is a direct demonstration of the development of anaphylaxis. One of the limitations of our study is that we did not measure other indicators of allergic mechanisms, such as IgG, which needs to be further improved in future studies.

We fully recognize the gap in the translation of animal model findings to practical applications. Therefore, a key design and core strength of this study is that we not only identified candidate biomarkers in rat models, but more importantly, we collected small but valuable real human forensic FA autopsy samples and performed preliminary validation of the key candidate (tryptophan). The potential forensic application of this study's findings is greatly enhanced by the observation of a significant trend in tryptophan levels in human FA samples, which is consistent with animal models, providing the most direct and strong supporting evidence for the translational potential of animal model findings.

In conclusion, we fully share the reviewer's concerns regarding the accuracy of the animal model. There is no animal model that can perfectly replicate the full complexity of human disease, especially when multiple pathologies are involved, such as FA overlapping cardiovascular disease. The main goal of this study was to explore unique and detectable metabolomic and gut microbiome profiles in FA, AMI, and superposition of both (CAA) states under tightly controlled experimental conditions, and to identify potential postmortem biomarkers. The models in our study are relatively recognized standard models in the respective fields (single FA or single AMI), and reliably induce the core pathophysiological processes of the target disease. The CAA model is an innovative attempt to simulate the key pathological interactions. These models allow for in-depth mechanistic exploration and biomarker discovery without confounders that are common in human cases, such as medication history and differences in time from death to autopsy, allowing us to identify disease-specific signals that may be masked in complex human settings.

Round 3

Reviewer 1 Report

Comments and Suggestions for Authors

The authors have thoroughly addressed my comments and concerns, provided a clear rationale for the use of the animal model, and appropriately acknowledged the study’s limitations. The overall quality of the manuscript has significantly improved, and I support its publication in its current form.